# Cost-effectiveness and budget impact analysis of siponimod in the treatment of secondary progressive multiple sclerosis in Italy

Paolo Angelo Cortesi[1,2], Ippazio Cosimo Antonazzo[1]*, Claudio Gasperini[3], Mihaela Nica[4], Daniela Ritrovato[4], Lorenzo Giovanni Mantovani[1,2]

1 Research Centre on Public Health (CESP), University of Milano-Bicocca, Monza, Italy, 2 Value-Based Healthcare Unit, IRCCS MultiMedica, Sesto San Giovanni, Italy, 3 Department of Neurology, Multiple Sclerosis Centre, San Camillo-Forlanini Hospital, Rome, Italy, 4 Novartis Farma SpA, Origgio, Italy

* ippazio.antonazzo@unimib.it

## Abstract

### Background

Siponimod is an effective treatment for patients with secondary progressive multiple sclerosis (SPMS), with active disease evidenced by relapses or imaging features characteristic of multiple sclerosis inflammatory activity, however there is a need to evaluate its economic value and sustainability compared to other disease modifying-therapies (DMTs).

### Objective

To estimate the siponimod cost-effectiveness profile and its relative budget impact compared with other DMTs, by using the Italian National Healthcare System perspective.

### Methods

We performed: 1) a cost-effectiveness analysis (CEA) vs interferon beta-1b using an analytical Markov model and a life time-horizon, and 2) a budget impact analysis by using 3-years time-horizon. The results were reported as incremental cost-effectiveness ratio (ICER) and net-monetary benefit (NMB) for CEA, using a willingness to pay threshold of €40,000 per QALY gained, and as difference in the overall budget (Euro) between the scenario with and without siponimod for budget impact.

### Results

In the base case scenario siponimod resulted cost-effective compared with interferon beta-1b 28,891€ per QALY. Overall, the market access of siponimod was associated to an increased budget of about 3€ millions (+0.9%) in the next 3 years simulated.

### Conclusion

Compared to interferon beta-1b, siponimod seems to be cost-effective in SPMS patients and sustainable, with less than 1% overall budget increased in the next 3 years. Future studies need to confirm our results in the real word setting and in other countries.

**Data Availability Statement:** All relevant data are within the manuscript and its Supporting Information files.

**Funding:** The study was supported by Novartis SpA in the form of a grant. There are no grant numbers to declare. Novartis also provided support in the form of salaries for MN and DR. The specific roles of these authors are articulated in the 'author contributions' section. The funders had no role in study design, data collection and analysis, decision to publish, or preparation of the manuscript.

**Competing interests:** The authors have read the journal's policy and have the following competing interests: MN and DR are employees of Novartis. PAC has received a research grant from Baxalta, now part of Shire, and speaking honoraria from Pfizer and Roche, outside the submitted work. LGM has received grants and personal fees from Bayer AG, Boehringer Ingelheim, Pfizer and Daiichi-Sankyo, outside the submitted work. CG has received fees as invited speaker or travel expenses for attending meetings from Biogen, Merck-Serono, Teva, Sanofi, Novartis, and Genzyme, outside the submitted work. The authors would like to declare the following marketed products associated with this research: Mayzent (Siponimod). This does not alter our adherence to PLOS ONE policies on sharing data and materials.

## Introduction

Multiple sclerosis (MS) is an immuno-mediated disease of the central nervous system that affects over 2.2 million of people worldwide [1]. It represents a leading cause of disability in young, mainly female, individuals. The relapsing remitting (RR) form is the most diagnosed type of MS, and about two-thirds of the RRMS patients transit to a more severe form, during the disease progression, called secondary progressive MS (SPMS) [2, 3]. SPMS is characterized by continuous accumulation of disability independent of the relapse.

The progression of the disease and its management has a significant impact on patients' and care givers' quality of life and it is typically associated to a high economic burden for payers and society [4]. Direct and indirect costs such as facilities access, informal care, and loss of productivity are strictly dependant on disease severity, with increased cost that parallel progression of disability [4–7]. In the last decades, several disease-modifying therapies (DMTs) have been marketed as treatment for RRMS. These therapies have improved clinical outcomes, reducing the disability progression and relapse rate, in treated patients. So far, few DMTs are available for the treatment of patients with SPMS. This raised the necessity to have more cost-effective therapeutic options to treat these patients.

Siponimod is a sphingosine-1-phosphate (SIP) receptor modulator specifically developed for patients with SPMS. The new drug is highly selective for SIP1 and SIP5 receptors which are involved in central nervous system homeostasis [8, 9]. Siponimod (Mayzent®), is an oral treatment, indicated as first-line of treatment for patients with SPMS who do not present the CYP2C9*3*3 genotype. The efficacy of siponimod has been demonstrated in a recent registrative study, a double-blind, placebo controlled phase 3 trial named EXPAND. In the aforementioned trial, siponimod significantly reduced disability progression rate in patients with SPMS (Hazard ratio: 0.74; 95Confidence intervals [95%CI] 0.60–0.92; risk reduction: 26%) compared with those treated with into the placebo group [10]. Regarding the safety profile of the new drug, in the EXPAND trial it showed a safety profile comparable to that observed for other DMTs [10].

After the EMA approval, the new drug has also been approved and reimbursed in Italy as a therapeutic option for patients with SPMS with active disease evidenced by relapses or imaging features characteristic of multiple sclerosis inflammatory activity. Considering the high economic and social costs associated to SPMS and the strong correlation by increasing level of disability and increasing socio-economic burden [5, 11], is fundamental for the National Health System (NHS) understanding the return in terms of health and economic outcomes from the possible investment in pharmacological treatment in early phase of SPMS. These need is particularly important in Italy when a universal health coverage is guarantee by the NHS and where a unique healthcare budget is generally distributed between the different patients based on their need and the cost and effectiveness of available interventions. Therefore, our study was designed to evaluate the cost-effectiveness of siponimod and its' budget impact versus other DMTs when used for treating patients with SPMS in Italy. In this study, we used the Italian NHS perspective.

## Materials and methods

The study included two main analyses: 1 –a cost-effectiveness analysis based on the development of a Markov model to assess the value of siponimod compared with interferon beta-1b; and 2 –a budget impact analysis based on the development of a budget impact model to assess the economic impact of siponimod on Italian NHS.

## Analysis 1: Cost-effectiveness analysis

A cohort-based multi-state Markov model was developed in Microsoft Excel to simulate the cost and effectiveness of treatment in patients with SPMS. The model simulated the natural history of SPMS based on three clinical events: disability progression, relapse and death. The model also included the impact of siponimod and other DMTs which are associated to reduction of disability progression and relapse rate with consequent improvement of both patient's management and their quality of life. The improvement or worsening of patients' disability was assessed by using the Expand Disability Status Scale (EDSS) score.

The EDSS score was used to define the health states, in particular, in this study each EDSS score is associated to one health state (Fig 1). At the beginning of the simulation, the SPMS patients reported an EDSS level within 3 and 6.5 (as per the registrative EXPAND trial) [10]. For each cycle of simulation patients could experience higher or lower EDSS level or alternatively could remain stable. In the model the rate of disability was strictly associated to EDSS level, while, mortality rate applied in the model was influenced by age, sex and EDSS level. For mortality, the model assumed an indirect effect of DMTs. Higher mortality risk is associated to higher EDSS level in the model; reducing the disability progression, siponimod reduce also the mortality risk.

In the model, patients with EDSS level ≤6.5 were assumed eligible for treatment. We assumed also that the treatment was stopped when patients reached an EDSS value of 7. The model included 10 states: 9 for each EDSS level and 1 corresponding to death. Additionally, for each EDSS level the model considered both treated and not treated patients conditions to account for treatment interruption. Within the model, the patients could move among higher or lower EDSS level, can discontinue or not the treatment, and can die.

In this cost-effectiveness analysis, siponimod was compared with interferon beta-1b. In Italy, this active substance is marketed in two formulation called Extavia® and Betaferon® that have same efficacy and cost. The data on drugs efficacy was retrieved from recent matching-adjusted indirect comparison meta-analysis [12]. In this study, efficacy of siponimod was compared with interferon beta-1a, interferon beta-1b and natalizumab. However, in our model we included only siponimod and interferon beta-1b because in Italy only these drugs were approved for the treatment of patients with SPMS. The efficacy data was retrieved by the Matched Adjusted Indirect Comparison (MAIC) published by Samjoo et al [12]. MAIC analyses provided an anchored indirect comparison due to the common comparator arm in each comparison (placebo) [13]. We used the relative effectiveness of interferon beta-1b and siponimod versus placebo in order to adjust the disability progression matrix estimated based on placebo arm of siponimod trial. Further, the relative efficacy estimated with the MAIC was based

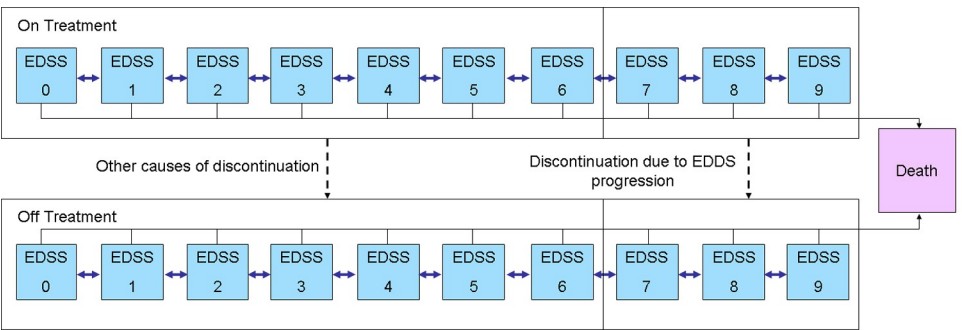

**Fig 1. Cost-effectiveness model structure.**

on the siponimod trial data and the interferon beta-1b trial conducted by Panitch et al [14]. The study by Panitch et al., conducted in North America, was selected because was the one that assessed the impact of treatment on disability progression using the confirmed disability progression at 6 months [14]. Further, we excluded ocrelizumab from the cost-effectiveness analysis because no specific and comparable data for SPMS population was available in the literature, as confirmed in the preliminary trial screening conducted by Samjoo et al. and their MAIC [12].

The analysis assumed a life-time horizon, 1-year cycle and a discount rate of 3% for both cost and benefits, as specified by Italian Medicine Agency (AIFA) guideline for economic evaluation of pharmaceuticals. In this simulation we adopted the perspective of Italian NHS.

Clinical, quality of life, management and relapse cost data input are reported in Table 1 [10, 14–18]. Table 2 reported DMTs efficacy and discontinuation risk. The DMTs efficacy was assessed using the Confirmed disability progression at 6 months (CDP-6) following the indication provided by the National Institute for Health and Care Excellence (NICE) for economic evaluation of DMTs in multiple sclerosis. In the multiple sclerosis DMTs assessment, NICE reported that CDP-6 is considered a more specific measure than at 3 months [19]. Treatment discontinuation included in the cost-effectiveness analysis includes withdrawal due to adverse events (AEs) or lack of effectiveness. Table 3 reported the DMTs ex-factory price considered for the analysis and their administration/monitoring costs [20]. More details on data input estimation are reported in the S1 Appendix.

**Outcome.** In the model the following parameters were estimated: cost in euros (€) of treatments, life-year (LYs), and quality-adjusted life-year (QALYs). The model results were combined to estimate the incremental cost-effectiveness ratio (ICERs) expressed as € per QALY gained. The ICER was computed comparing siponimod with the interferon beta-1b. Additionally, the value of siponimod was quantified by calculating the incremental net monetary benefit (NMB) using a willingness to pay (WTP) of €40,000 per QALY gained.

**Sensitivity analysis.** To assess uncertainty around the model parameters, a one-way sensitivity analysis (OWSA) and a probabilistic sensitivity analysis (PSA) were conducted. The OWSA was performed to assess the impact of each parameter variation on the model NMB estimated. The PSA was performed assessing the impact of all parameters' variability. The results from the PSA were used to generate a cost-effectiveness acceptability curve (CEAC) to explore the probability of each treatment strategy to be economically attractive. The CEAC indicates the probability that each treatment has to be cost-effective, given the values and uncertainty of the parameters used in the model and for different values of the acceptable WTP. Finally, an alternative scenario analysis based on DMTs efficacy assessed using the Confirmed disability progression at 3 months (CDP-3) was performed. The CDP-3 data was retrieved by the MAIC published by Samjoo et al [12]. with an Hazard Ratio of 0.74 (95%CI: 0.60–0.91) for interferon beta-1b and 0.61 (0.32–1.16) for siponimod.

## Analysis 2: Budget impact analysis

The budget impact analysis was performed to assess the impact of siponimod use in a cohort of SPMS patients in the Italian market. The model compared two scenarios according to siponimod presence on the market: "Scenario no-Sipo" where the study drug was not present and "Scenario Sipo" which includes siponimod as possible treatment on the market.

The model was based on epidemiological data of SPMS in Italy (Table 4) and on DMTs treatment utilization (Table B in S1 Appendix) [18, 21–24]. In the model, we included prevalent SPMS population with EDSS score ranged 3.0 to 6.5. In the analysis, the number of

**Table 1. Clinical, quality of life and management cost data input.**

| Parameters | Value | | | | | | | | | | Reference |
|---|---|---|---|---|---|---|---|---|---|---|---|
| EDSS | 0 | 1 | 2 | 3 | 4 | 5 | 6 | 7 | 8 | 9 | |
| **Cohort characteristics** | | | | | | | | | | | |
| Age, mean (years) | 48.0 | | | | | | | | | | 10 |
| Male (%) | 39.9 | | | | | | | | | | 10 |
| Disability distribution (%) | 0.00 | 0.00 | 0.49 | 9.32 | 18.59 | 16.09 | 55.33 | 0.18 | 0.00 | 0.00 | 10 |
| **Clinical data, annual probability** | | | | | | | | | | | |
| Relapse | 0.000 | 0.000 | 0.465 | 0.161 | 0.218 | 0.168 | 0.126 | 0.276 | 0.276 | 0.276 | 10,14,15,16 |
| Mortality rate | 1.00 (0.80–1.20) | 1.43 (1.14–1.72) | 1.60 (1.28–1.92) | 1.64 (1.31–1.97) | 1.67 (1.34–2.00) | 1.84 (1.47–2.21) | 2.27 (1.82–2.72) | 3.10 (2.48–3.72) | 4.45 (3.56–5.34) | 6.45 (5.16–7.74) | 17 |
| **Utility** | | | | | | | | | | | |
| Mean (range) | 0.832 (0,646–0,957) | 0.791 (0,620–0,920) | 0.737 (0,583–0,866) | 0.651 (0,520–0,771) | 0.582 (0,467–0,693) | 0.501 (0,403–0,598) | 0.412 (0,333–0,494) | 0.300 (0,243–0,360) | -0.041 (-0,033 - -0,049) | -0.214 (-0,174 - -0,257) | 10,11 |
| **Cost, mean (range)** | | | | | | | | | | | |
| Overall management | € 2.102 (1.720–2.543) | € 2.102 (1.720–2.543) | € 2.102 (1.720–2.543) | € 2.102 (1.720–2.543) | € 4.822 (3.946–5.834) | € 4.822 (3.946–5.834) | € 4.822 (3.946–5.834) | € 8.052 (6.589–9.742) | € 8.052 (6.589–9.742) | € 8.052 (6.589–9.742) | 18 |
| Relapse | € 405 | | | | | | | | | | 18 |

subjects treated with the study drug increased over the observed period (3 years time-horizon) (Fig A in S1 Appendix).

The model included: the DMTs costs, their administration/monitoring costs and AEs costs (Table 3 and Table A in S1 Appendix). In addition to the cost-effectiveness analysis, the budget impact included an additional DMT in the analysis: ocrelizumab. Ocrelizumab was considered in the budget impact analysis based on its approval for RMS form of MS, which includes SPMS relapsing patients as well. In the budget impact analysis, the cost of ocrelizumab was estimated using a regimen of 600 mg every 6 month and an ex-factory cost of € 6.250,0 per 330 mg [20]. The yearly administration/monitoring costs of ocrelizuamb was € 1,150.0 in the first year and € 363.0 after the second year [18]. The AEs management cost was estimated based on a previous economic evaluation conducted in Italy [18].

Moreover, the model included the costs associated to management of patients and the relapses from the Italian NHS perspective (Table 1). The detailed data input used in the analyses are reported in the S1 Appendix.

**Outcome.** Te model estimated the annual cost per patient for each type of treatment by using the parameters costs and healthcare resources consumption. These costs were associated to the epidemiological and market share data to estimate the 3 years overall cost of the scenario with and without siponimod. The budget impact of siponimod in Italy, with a 3-years time horizon, was the result of the cost difference between the two scenarios.

**Table 2. Disease modifying therapies efficacy and discontinuation.**

| DMT | ARR* | | CDP-6 months* | | Discontinuation§ | | Reference |
|---|---|---|---|---|---|---|---|
| | RR | 95%CI | HR | 95%CI | HR | 95%CI | 12 |
| Interferon beta | 0.65 | 0.48–0.88 | 0.93 | 0.71–1.20 | - | | |
| Siponimod | 0.59 | 0.35–0.99 | 0.50 | 0.35–0.74 | 0.87 | 0.64–1.18 | |

ARR = annual relapse rate; CDP = confirmed disability progression; HR = Hazard ratio; RR = relative risk; SE Standard error.

* Drug compared with placebo; § Siponimod versus interferon beta 1b.

**Table 3. Disease-modifying therapies (DMTs) costs.**

| DMT | Dose | Unit per pack | List price (€) | Price ex-factory | Administration and monitoring costs | Reference |
|---|---|---|---|---|---|---|
| **INF bera-1b (Extavia®)** | 0,25 mg every 48 hours | 5 vials | € 470.9 | € 285.3 | € 1,137 first year € 412 after first year | 20 |
| **INF bera-1b (Betaferon®)** | 0,25 mg every 48 hours | 15 vials | € 1412.8 | € 856.01 | | |
| **Siponimod** | 2 mg/day | 28 tablets | € 3,120.7 | € 1,890.9 | € 1,272 first year € 309 after first year | 20 |

## Results

### Cost effectiveness analysis

As reported in Table 5, siponimod resulted the most effective treatment (1.05 QALY gained) but also more expensive (€30.308 per patient) compared with interferon beta-1b. The increased efficacy and costs associated with siponimod resulted in an ICER of € 28,891 per QALY gained, and a NMB of € 11,654 using a willingness to pay of € 40,000 per QALY gained.

The one-way sensitivity analysis confirmed the good cost-effectiveness profile of siponimod compared with interferon beta-1b. In the OWSA, the treatment efficacy on disease progression confirmed at 6-months was the parameter that mostly affected the results (Fig 2). The impact of treatment efficacy was also confirmed by the alterative scenario analysis. This analysis based on CDP-3 data, siponimod reported an ICER of € 80,063 compare to interferon beta-1b.

PSA results confirmed the good cost-effectiveness profile of siponimod compared to interferon beta-1b, with siponimod that reported a 78% of probability to be the cost-effective treatment option at a WTP threshold of 40,000 euros (Fig 3).

### Budget impact analysis

Based on the model assumptions, the estimated target population with SPMS was composed by 5827 patients.

Fig 4 shows the number of patients treated with each DMT in the observed period. In the scenario with siponimod, the number of patients potentially treated with the new treatment increased over time, from 405 during the first year up to 2,236 in the third year. As reported in the figure, with the introduction of siponimod, the number of patients with SPMS treated with interferon beta-1b decreased mainly in the first 2 years, while patients treated with ocrelizu-mab decreased mainly in the last year.

**Table 4. Epidemiological data on study population.**

| Variable | Value | Source |
|---|---|---|
| Italian Population | 59,641,488 | 21 |
| Multiple sclerosis (MS) prevalence rate | 0.2% | 22 |
| Number of prevalent MS patients in the model | 119,283 | *Estimated* |
| SPMS prevalence | 13.7% | 18, 23 *and expert opinion* |
| Number of subjects with SPMS | 16,342 | *Estimated* |
| Percentage of patients with SPMS age 18–60 and EDSS between 3–6.5 | 91.8% | 18, 23 *and expert opinion* |
| Number of patients with SPMS age 18–60 and EDSS between 3–6.5 | 15,001 | Estimated |
| Percentage of patients with active SPMS | 60.0% | 23 |
| Number of patients with active SPMS | 9,002 | *Estimated* |
| Percentage of patients with active SPMS and under-treatment | 65.0% | 21 |
| Number of patients with active SPMS and under-treatment | 5,851 | *Estimated* |
| Percentage of patients eligible for siponimod treatment | 99.6% | 24 |
| Number of patients eligible for siponimod treatment | 5,827 | *Estimated* |

**Table 5. Cost-effectiveness analysis results.**

| DMTs | Costs (€) | Δ Costs (€) | LYs | ΔLYs | QALYs | ΔQALYs | ICER (€ per QALY gained) | NMB (WTP €40,000 per QALY) |
|---|---|---|---|---|---|---|---|---|
| Interferon beta-1b | 152,435 | | 17.77 | | 4.44 | | | |
| Siponimod | 182,744 | 30.308 | 18.05 | 0.28 | 5.49 | 1.05 | 28,891 | 11,654 |

ICER: Incremental cost-effectiveness ratio; LY = life years; QALYs = Quality Adjusted Life Years; NMB = Net Monetary Benefit with a willingness to pay (WTP) of €40,000 per QALY gained.

The economic impact of siponimod was estimated in an increase of 2,819,026 million (0.9%) in 3 years simulated, with an incremental cost of 1.1% (1,214,249 millions) in the first year, 1,6% (1,760,975 millions) in the second year and -0,14% (-156,198 thousands) in the last year (Fig 4).

## Discussion

This study attempts the cost-effectiveness and sustainability of siponimod compared with other DMTs as potential treatment for SPMS patients in an Italian setting.

In the cost-effectiveness analysis, siponimod was compared with interferon beta-1b. In the base case scenario, siponimod resulted cost-effective compared with the comparator. In particular, we found an ICER per QALY gained of € 28,891 compared with the interferon beta-1b and a NMB of € 11,654 considering a WTP of € 40,000. These results were mostly affected by DMTs efficacy, as showed by sensitivity analysis and alternative scenario analysis. Our analysis suggests that the introduction of siponimod in the Italian market was associated with an increased budget expenditure of about 3 millions of euros in a three years time horizon.

To the best of our knowledge, our study is the first analysis aimed at assessing both the cost-effectiveness and budget impact of siponimod in Italy. Our cost-effectiveness findings is in contrast with a previous one conducted in the USA [25]. In the aforementioned study,

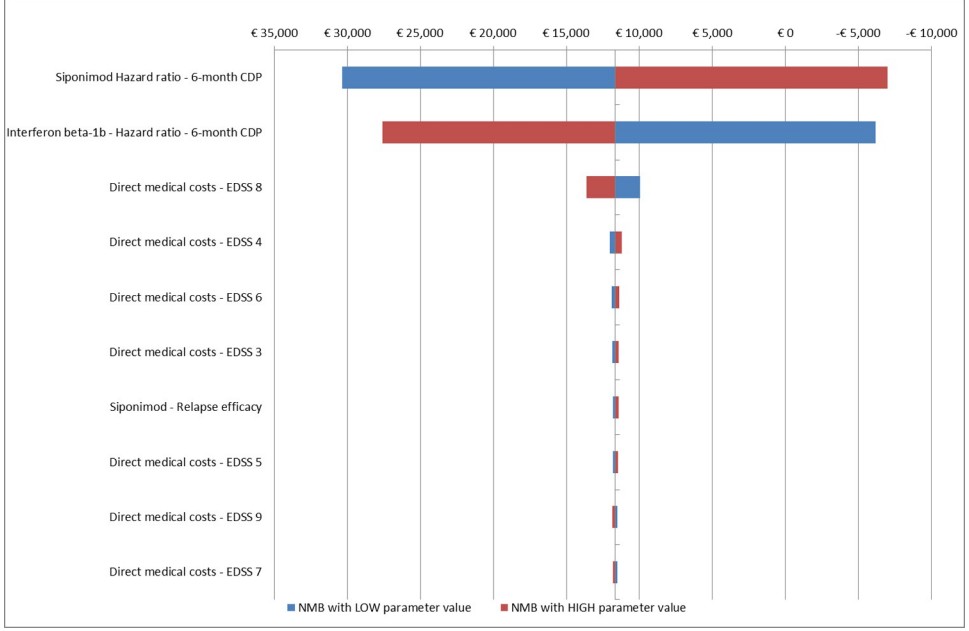

**Fig 2. One way sensitivity analysis of cost effectiveness of siponimod compared with Interferon beta-1b.**

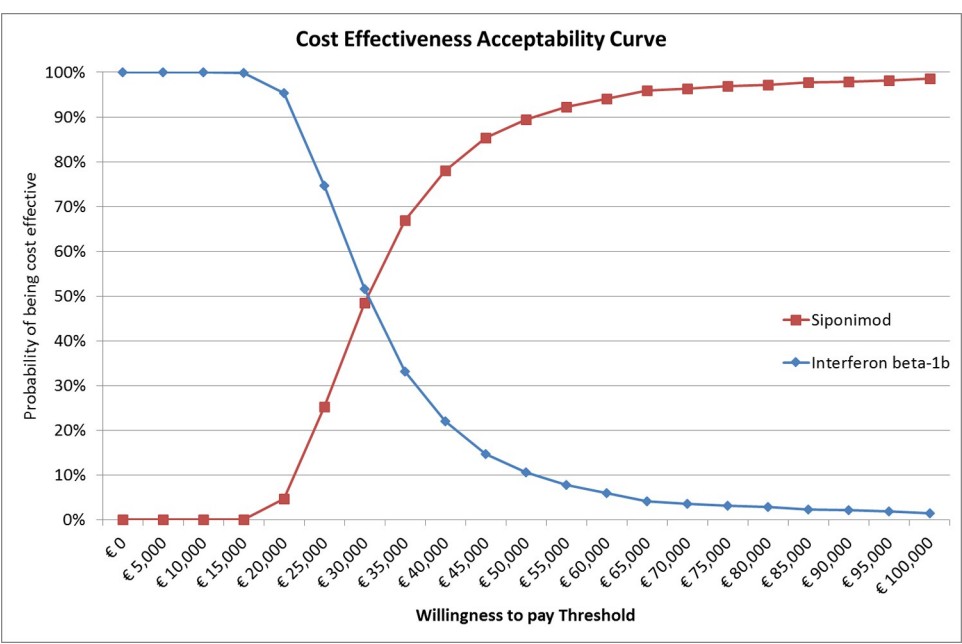

**Fig 3. PSA analysis of siponimod compared with interferon beta-1b.**

siponimod resulted not cost-effective compared with the best supportive care, as estimated by the placebo arm of the pivotal trial, reporting an additional cost-per-QALY gained of $ 433,000 in active SPMS population. The analysis conducted for USA setting reported also the cost-effectiveness results of a comparison between siponimod and interferon beta-1b based on a MAIC. The results of this analysis were even worse with an ICER of € $2.11 million per QALY gained; however, no details on the comparative efficacy values used for the two treatments were reported in the study. The lack of these data make impossible to compare findings from the USA study with those from our study. The value of Siponimod was also assessed in Canada and UK [26, 27]. Canada reported a need of siponimod price cut to be considered cost-effective compare to best supportive care, while in the UK, siponimod was considered cost-effective considering the limited alternative treatment options for this population and

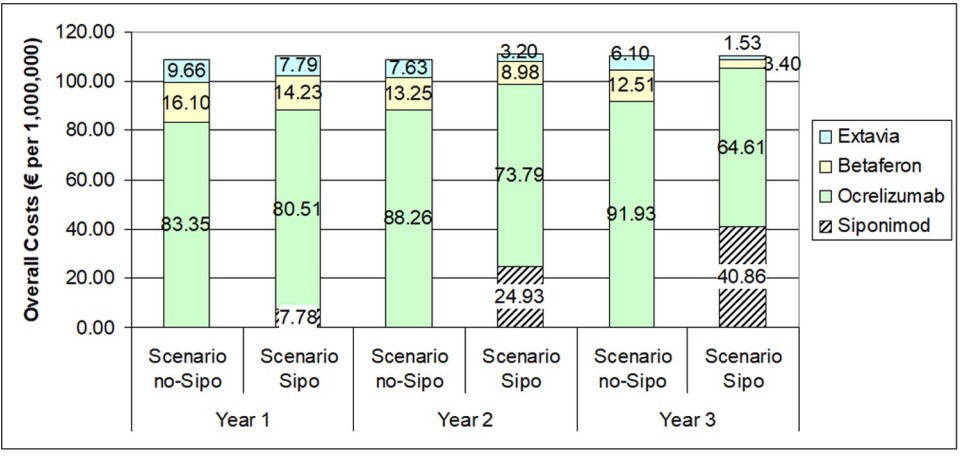

**Fig 4.**

considering that the cost-effectiveness estimates were within the range that NICE normally considers an acceptable use of NHS resources. As highlighted by the results of these studies, the variability of economic results due to the different treatments price and healthcare system required specific analysis for each country. Our results might contribute to fill this gap and provide information on the value of siponimod in a country as Italy, which could be considered as reference for other European countries. In our analysis, we used an active comparator (interferon beta-1b) reimbursed and indicated as treatment for SPMS patients in Italy, with an ICER of € 28,891 per QALY gained that was under the Italian threshold of € 40,000. Therefore, based on our study results, siponimod can be considered a cost-effective treatment for SPMS patients compare to interfere beta-1b in Italy [28].

However, has recognized in the recent years, the value showed in the cost-effectiveness results must be considered in light of the relative sustainability of the introduction of new intervention [29]. Our study provided a complete picture of the value and sustainability of siponimod in Italy, including information on both cost-effectiveness and the budget impact of siponimod for the Italian NHS perspective. In the model a time horizon of 3 years was used to explore the impact of the new treatment in terms of monetary impact and potentially treated individuals. In our analysis, we assumed the number of subjects treated with siponimod increased overtime with consequent reduction of those treated with interferons and ocrelizumab in favour of siponimod. As a result, the introduction of siponimod increased expenditure of about 3 million (+0.9% of total budget) over 3 year of observation.

In this regard, two important aspects should be emphasised. First, the number of therapies available for SPMS is still scarce, therefore it is essential to expand the drug armamentarium in order to guarantee the best treatment to all patients (i.e, in case of patients who are intolerant to other DMTs or in case of inefficacy of available treatment). Interferon beta-1b has been for long time the only treatment licensed for secondary progressive multiple sclerosis with active disease evidenced by relapses in Italy. Recently, ocrelizumab was approved by the Italian Medicines Agency (AIFA) for the treatment of adult patients with relapsing forms of multiple sclerosis (RMS) with active disease defined by clinical or imaging features (GU Serie Generale n.204 del 03-09-2018), giving a second option to treat secondary form with relapse. Second, as reported in the cost-effectiveness analysis, the advantages associated with the use of siponimod compared with interferon beta-1b might justify the difference in treatment costs between the two products. Further, siponimod is associated with a small impact on Italian NHS budget, making this treatment option a sustainable one for the system. Even if interferon beta-1b had a lower cost than siponimod, the difference in the budget impact was small due to the availability of a new treatment option for SPMS with relapse in Italy, ocrelizumab that was included in our analysis. Ocrelizumab has not been included in the cost-effectiveness analysis due to the lack of efficacy data on a comparable population with those included in the siponimod trial, however it is actually a prescribe treatment for these patients with a higher price of interferon beta-1b. Including the possible use of siponimod instead of interferon beta-1b an ocrelizumab made the analysis more reliable and in line with the real word of Italian setting. The results of the analysis provide a complete picture of siponimod impact and help Italian healthcare decision makers to define the implementation of this treatment in MS centres.

These results should be interpreted taking into account some limitations, including the variety of sources of data that was used to identify inputs for model's parameters. The comparative efficacy of siponimod and interferon beta-1b was retrieved from a MAIC published by Samjoo et al [12]. Even if MAIC is the best approach to estimate the relative efficacy between siponimod and interfern beta-1b, some limitations are associated to this approach. As reported by Samjoo et al., these limitations include differences in trial design and patient characteristics, which were not fully adjusted due to a paucity of data [12]. Inclusion criteria for the IFNb-1b

trials were broader than EXPAND thus precluding our ability to align on all variables despite individual patient data from EXPAND. Fortunately, for the key treatment effect modifiers identified by clinical experts, multivariable adjustments in the MAIC were possible. The definition of "disability progression" reported difference between studies. Although these MAICs adjust for observed baseline differences between siponimod and comparator trials, they are comparisons of randomized treatment groups and may therefore be biased by potential unobserved cross-trial differences. In addition, it should be recognized that the follow-up of clinical trials is shorter than time horizon included in the analysis, therefore our analysis may overestimate the value of the therapy if treatment efficacy decline overtime. Hence, future studies need to update this data with longer follow-up to understand the medium, and long-term relative efficacy of the studied treatment. The population of the EXPAND trials is representative of the UK population. Although, the Italian population can be considered similar to UK population in terms of population characteristics and prevalence of disease [1], this cannot be the same for other countries. Finally, ocrelizumab was not included in the cost-effectiveness analysis. This choice was made given that the trials for SPMS patient population were only available for siponimod and interferon beta-1b. The pivotal trial of ocrelizumab (OPERA) was done on RMS patients that are a mix of RRMS and relapsing SPMS patients [30]. The differences of the patient population included in the OPERA and EXPAND trials make the efficacy data of the two treatments not comparable and thus not applicable in a cost-effectiveness analysis. Additional evidence is required to perform a reliable cost-effectiveness analysis of siponimod vs ocrelizumab.

## Conclusion

Siponimod represents a new first line treatment for patients with SPMS. This study provides information to assess the value of siponimod for patients with SPMS, indicating it as a cost-effective treatment option compared to interferon beta-1b, with a low impact on the healthcare budget (+0.9%) over 3 years of observation. This findings represent a valuable evidence that can be used by healthcare decision makers and clinicians to implement the use of siponimod for treating SPMS patients in clinical practice. So far, this is the first evidence on this topic in Italy; further studies, with a longer follow up period, may be useful to confirm our findings in the real world setting.

## Supporting information

**S1 Appendix. Data input details for cost-effectiveness and budget impact analysis.** (DOC)

## Author Contributions

**Conceptualization:** Paolo Angelo Cortesi, Mihaela Nica, Daniela Ritrovato, Lorenzo Giovanni Mantovani.

**Data curation:** Paolo Angelo Cortesi, Ippazio Cosimo Antonazzo, Mihaela Nica.

**Formal analysis:** Paolo Angelo Cortesi, Ippazio Cosimo Antonazzo, Mihaela Nica, Daniela Ritrovato.

**Funding acquisition:** Mihaela Nica, Daniela Ritrovato.

**Methodology:** Paolo Angelo Cortesi, Claudio Gasperini.

**Project administration:** Mihaela Nica, Daniela Ritrovato.

**Resources:** Daniela Ritrovato, Lorenzo Giovanni Mantovani.

**Supervision:** Paolo Angelo Cortesi, Lorenzo Giovanni Mantovani.

**Validation:** Claudio Gasperini.

**Visualization:** Ippazio Cosimo Antonazzo, Mihaela Nica, Lorenzo Giovanni Mantovani.

**Writing – original draft:** Paolo Angelo Cortesi, Ippazio Cosimo Antonazzo, Mihaela Nica, Daniela Ritrovato, Lorenzo Giovanni Mantovani.

**Writing – review & editing:** Paolo Angelo Cortesi, Ippazio Cosimo Antonazzo, Claudio Gasperini, Mihaela Nica, Daniela Ritrovato, Lorenzo Giovanni Mantovani.

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
