## [Decision Letter · Decision Letter 0]

8 Oct 2021

PONE-D-21-28328Cost-effectiveness and budget impact analysis of Siponimod in the treatment of Secondary progressive Multiple Sclerosis in ItalyPLOS ONE

Dear Dr. Antonazzo,

Thank you for submitting your manuscript to PLOS ONE. After careful consideration, we feel that it has merit but does not fully meet PLOS ONE’s publication criteria as it currently stands. Therefore, we invite you to submit a revised version of the manuscript that addresses the points raised during the review process.

We look forward to receiving your revised manuscript.

Kind regards,

Marcello Moccia

Academic Editor

PLOS ONE

Journal Requirements:

[PAC has received a research grant from Baxalta now part of Shire, and speaking honoraria from Pfizer and Roche. LGM has received grants and personal fees from Bayer AG, Boehringer Ingelheim, Pfizer and Daiichi-Sankyo. ICA has no conflicts of interest to disclose. MN and DR are employes of Novartis Pharma AG.  CG has recieved fees as invited speaker or travel expenses for attending meeting from Biogen, Merck-Serono, Teva, Sanofi, Novartis and Genzyme.] 

6. Please ensure that you refer to Figure 2 in your text as, if accepted, production will need this reference to link the reader to the figure.

Reviewers' comments:

Reviewer's Responses to Questions

**Comments to the Author**

1. Is the manuscript technically sound, and do the data support the conclusions?

Reviewer #1: Yes

Reviewer #2: Partly

2. Has the statistical analysis been performed appropriately and rigorously? 

Reviewer #1: No

Reviewer #2: I Don't Know

3. Have the authors made all data underlying the findings in their manuscript fully available?

Reviewer #1: Yes

Reviewer #2: No

4. Is the manuscript presented in an intelligible fashion and written in standard English?

Reviewer #1: Yes

Reviewer #2: Yes

5. Review Comments to the Author

Reviewer #1: The submitted study is specifically focused on the pharmacoeconomic and economic characteristics of Siponimod from the perspective of the Italian NHS. The authors have presented in a comprehensive and clear style most of the methods and materials for CEA and BIA. Both CEA and BIA are performed according to the approved and accepted methodology.

The discussion and introduction section could be improved. Deeper analysis/discussion of the results and their importance and value for the NHS and patients could be done. Additional explanations regarding the statistical methods which were applied are needed.

Reviewer #2: The EMA and FDA indications for siponimod are for active SPMS (i.e. relapsing or inflammation on MRI). It remains unclear whether siponimod reduces progression independent of its effect on relapse reduction. This is important nuance which has not been described in the manuscript. In fact you say “While only siponimod have showed a statistically significant modification of natural history by reducing progression…” which is misleading.

Abstract

-State which threshold you are applying to determine if the drug is cost-effective

Introduction

-This section sets up a good context and rationale for the analysis. Please be precise about how you describe the indication (active SPMS) unless that distinction isn't relevant in Italy.

-Anyone familiar with EXPAND will notice that you are reporting the secondary endpoint from the trial (CDP-6) rather than the primary endpoint (CDP-3). While the results were of similar magnitude, citing the slightly more favorable secondary endpoint over the primary will make the paper appear biased unless this decision is justified in text.

Analysis 1

-Please clarify what caused treatment interruptions

-State whether you used list or ex-factory prices for drug costs. Also, you state that Extavia and Betaferon have the same costs but that doesn’t appear to be true of the ex-factory costs reported in Table 3

-Improvement in EDSS assumption: This likely only occurs in early stages of SPMS when recovering from a relapse—it is doubtful whether it would persist past a 1-year cycles or occur once patients progress to higher EDSS scores. An MS specialist may be able to advise on this matter.

-Table 1: please provide uncertainty intervals for the utilities

-Utilities associated with EDSS 8 and 9: Describe why you assumed a quality of life worse than death rather than 0

-Utilities: Describe how they were assessed (e.g., which instrument was used)

-Describe the interferon beta 1b trial from which the MAIC was conducted. It seems like you used the data from the North American trial. Please justify why you used this and consider running an additional analysis using the European trial data.

-The MAIC relied on summary data from the IFN trials and had a lot of limitations. For example individual trials used different definitions of CDP (including the North American trial) and the MAIC could not adjust for all potential effect modifiers. These limitations should be addressed in the discussion of your manuscript. Given these limitations, you may also want to consider adding an additional comparator for best supportive care (estimated by the placebo arm of EXPAND).

-Please clarify how you estimated the clinical effectiveness in the model. I’m not sure if I am understanding Table 2 correctly but it seems like you used the relative effects vs placebo rather than the indirect estimate from the MAIC of siponimod vs. IFN?

Results

-I wonder if the incremental cost is reported correctly. The ICER I calculate using the disaggregated costs is closer to what you report than the ICER I calculate when I use the incremental values from Table 5.

Discussion

-In the comparison to the US model, it would be better to review the full HTA report from the Institute for Clinical and Economic Review (rather than the summary in JMCP). They conducted a scenario analysis using the MAIC data to compare siponimod with IFN. You will also find a budget impact analysis in that report.

6. PLOS authors have the option to publish the peer review history of their article (what does this mean?). If published, this will include your full peer review and any attached files.

Reviewer #1: No

Reviewer #2: No

---

## [Author Response · Author response to Decision Letter 0]

27 Nov 2021

Dear Editor, 

Thanks for the opportunity to address the reviewers’ comments, which we have carefully considered when preparing the revised version of the manuscript. In the new version of manuscript, we have added more details in the main text and addressed the concerns raised by reviewers. 

Our point-by point responses are provided below

Reviewers' comments:

Reviewer #1: 

The submitted study is specifically focused on the pharmacoeconomic and economic characteristics of Siponimod from the perspective of the Italian NHS. 

The authors have presented in a comprehensive and clear style most of the methods and materials for CEA and BIA. Both CEA and BIA are performed according to the approved and accepted methodology.

1. The discussion and introduction section could be improved. Deeper analysis/discussion of the results and their importance and value for the NHS and patients could be done. 

We thank the reviewer for the comments. We have discussed more in details the importance of our results and the value for NHS and patients.

2. Additional explanations regarding the statistical methods which were applied are needed.

We have added in the text and in the supplementary material, more details regarding the statistical methods applied.

Reviewer #2: 

1. The EMA and FDA indications for siponimod are for active SPMS (i.e. relapsing or inflammation on MRI). It remains unclear whether siponimod reduces progression independent of its effect on relapse reduction. This is important nuance which has not been described in the manuscript. In fact, you say “While only siponimod have showed a statistically significant modification of natural history by reducing progression…” which is misleading.

We thank the reviewer for the comment. Our model and analysis are based on the data provided by the phase 3 clinical trial conducted by Kappos and colleagues were they showed a statistical significant reduction of the risk of disability progression using siponimod compare to placebo (Kappos 2018). This trial provided the data used for estimating the efficacy of siponimod in reducing the disability progression in our analysis.

While the statement “While only siponimod have showed a statistically significant modification of natural history by reducing progression…” could be considered misleading we have changed it reporting this new sentence in the article “While only siponimod have showed, in a randomized clinical trial, the possibility to reduce the risk of disability progression…”

Kappos L, Bar-Or A, Cree BAC, Fox RJ, Giovannoni G, Gold R, Vermersch P, Arnold DL, Arnould S, Scherz T, Wolf C, Wallström E, Dahlke F; EXPAND Clinical Investigators. Siponimod versus placebo in secondary progressive multiple sclerosis (EXPAND): a double-blind, randomised, phase 3 study. Lancet. 2018 Mar 31;391(10127):1263-1273. doi: 10.1016/S0140-6736(18)30475-6. Epub 2018 Mar 23. Erratum in: Lancet. 2018 Nov 17;392(10160):2170.

Abstract

2. State which threshold you are applying to determine if the drug is cost-effective.

We have added the threshold applied in the analysis.

Introduction

3. This section sets up a good context and rationale for the analysis. Please be precise about how you describe the indication (active SPMS) unless that distinction isn't relevant in Italy.

We thank the reviewer for the comment. We have better described the indication for SPMS.

4. Anyone familiar with EXPAND will notice that you are reporting the secondary endpoint from the trial (CDP-6) rather than the primary endpoint (CDP-3). While the results were of similar magnitude, citing the slightly more favorable secondary endpoint over the primary will make the paper appear biased unless this decision is justified in text.

We thank the reviewer for the comment. The choice of using the CDP-6 is related to the indication provided by the National Institute for Health and Care Excellence (NICE) for economic evaluation of DMTs in multiple sclerosis. In the ocrelizumab assessment, NICE reported that that confirmed disability progression at 6 months is considered a more specific measure than at 3 months. (https://www.nice.org.uk/guidance/ta533/resources/ocrelizumab-for-treating-relapsingremitting-multiple-sclerosis-pdf-82606899260869 )

We have explained this choice in the text and included the NICE reference.

Analysis 1

5. Please clarify what caused treatment interruptions.

We thank the reviewer for the comments. We have clarify on the article that treatment discontinuation considered in the cost-effectiveness model includes withdrawal due to adverse events (AEs) or lack of effectiveness.

6. State whether you used list or ex-factory prices for drug costs. Also, you state that Extavia and Betaferon have the same costs but that doesn’t appear to be true of the ex-factory costs reported in Table 3.

We thank the reviewer for the comment. We have stated in the main test that we used ex-factory price. We also have corrected a typo in the ex-factory price of Extavia, now also the ex-factory prices of both interferons are the same.

7. Improvement in EDSS assumption: This likely only occurs in early stages of SPMS when recovering from a relapse—it is doubtful whether it would persist past a 1-year cycles or occur once patients progress to higher EDSS scores. An MS specialist may be able to advise on this matter.

We thank the reviewer for the comment. We agree with the reviewers that EDSS improvement is associated to recovering from relapse, however the transition matrix and the relative probability of improvement and worsening of EDSS are based on statical methods approved and suggested in the literature and by the HTA national agency. This method is multi-state model (MSM) approach (using the “MSM” package in R) that give us the possibility to estimate the specific transition probability from each EDSS level to others. This approach include all EDSS variations avilble for each patients. The results obtained with this method have showed the possibility to move from higher to lower EDSS also for SPMS and PPMS (Fornari 2020, Palace 2014)

We have added a better explanation of the method used to estimate transition probability in the supplementary material. 

“Transition probabilities between EDSS states were estimated based on the data from placebo arm of EXPAND trial [Kappos]. A multi-state model (MSM) approach (using the “MSM” package in R) was applied to produce the transition probability matrix, following the approach used in the natalizumab assessment conducted by NICE (TA127) [NICE, 2007]. A MSM was fitted using information on the EDSS level recorded at each visit scheduled during the trial, the time spent in each EDSS state and the initial values of the transition intensity matrix. When the sample size was not big enough to calculate transition probability, data from the London Ontario MS dataset [Mauskopf 2016] was used. The transition probability matrix estimated with te MSM approach was validated comparing the model results after 1 and 2 years with the patient distribution by EDSS state observed in the EXPAND trial.” 

Palace J, Bregenzer T, Tremlett H, Oger J, Zhu F, Boggild M, Duddy M, Dobson C. UK multiple sclerosis risk-sharing scheme: a new natural history dataset and an improved Markov model. BMJ Open. 2014 Jan 17;4(1):e004073. doi: 10.1136/bmjopen-2013-004073. Erratum in: BMJ Open. 2014;4(1):e004073corr1. 

Fornari C, Cortesi PA, Capra R, Cozzolino P, Patti F, Mantovani LG. The disability progression of multiple sclerosis. Value in Health Volume 23 supplement 2, S624, December 01, 2020

National Institute for Health and Care Excellence (NICE). Natalizumab for the treatment of adults with highly active relapsingremitting multiple sclerosis; 2007. https://www.nice.org.uk/guida nce/ta127/history (Accessed 20 Oct 2021).

Mauskopf J, Fay M, Iyer R, Sarda S, Livingston T. Cost-efectiveness of delayed-release dimethyl fumarate for the treatment of relapsing forms of multiple sclerosis in the United States. J Med Econ. 2016;19(4):432–42

8. Table 1: please provide uncertainty intervals for the utilities.

We have added the intervals.

9. Utilities associated with EDSS 8 and 9: Describe why you assumed a quality of life worse than death rather than 0.

The utility applied to each EDSS level was the ones published for the economic evaluation of DMTs in the multiple sclerosis patient in Italy and estimated from EQ-5D data (Mantovani 2019)

The utility estimated with the EQ-5D questionnaire can be negative due to the fact that some health states described by the questionnaire has been considered worse than death by the representative sample of the general population included in the utility set generation studies (Dolan 1996, Scalone 2013).

In the literature is well establish the relationship between high EDSS level and negative utility value estimated with the EQ-5D (The EuroQol Group. https://euroqol.org/) (Kobelt 2017; Kobelt 2006).

We have better explained the source of utility data in the supplementary material.

Dolan P, Gudex C, Kind P, et al. The time trade-off method: Results from a general population study. Health Econ 1996; 5: 141–154

Scalone L, Cortesi PA, Ciampichini R, et al. Italian population-based values of EQ-5D health states. Value Health 2013; 16: 814–822

Lorenzo Giovanni Mantovani, Gianluca Furneri, Rossella Bitonti, Paolo Cortesi, Elisa Puma, Laura Santoni, Luca Prosperini . Cost-Effectiveness Analysis of Dimethyl Fumarate in the Treatment of Relapsing Remitting Multiple Sclerosis: An Italian Societal Perspective. Farmeconomia. Health economics and therapeutic pathways 2019; 20(1): 73-86

Kobelt G, Thompson A, Berg J, Gannedahl M, Eriksson J; MSCOI Study Group; European Multiple Sclerosis Platform. New insights into the burden and costs of multiple sclerosis in Europe. Mult Scler. 2017 Jul;23(8):1123-1136

Kobelt G, Berg J, Lindgren P, et al. Costs and quality of life of patients with multiple sclerosis in Europe. J Neurol Neurosurg Psychiatry 2006; 77: 918–926

10. Utilities: Describe how they were assessed (e.g., which instrument was used).

The utility data used in the analysis were retrieved by a recent economic evaluation of DMTs in the multiple sclerosis patient in Italy (Mantovani 2019) In that study, utility values from EQ-5D data weights for EDSS states without a relapse and during a relapse were derived from the delayed-release dimethyl fumarate clinical trial data by pooling observations for each EDSS state (0–9) and calculating the mean EuroQol EQ-5D index score for each state. This utility value were adjusted for the disutility associated to SPMS obtained from a survey conducted in the UK (Orme 2007). 

Mantovani LG, Furneri G, Bitonti R, Cortesi P, Puma E, Santoni L, Prosperini L. Cost-Effectiveness Analysis of Dimethyl Fumarate in the Treatment of Relapsing Remitting Multiple Sclerosis: An Italian Societal Perspective. Farmeconomia. Health economics and therapeutic pathways 2019; 20(1): 73-86

Orme M, Kerrigan J, Tyas D, et al. The effect of disease, functional status, and relapses on the utility of people with multiple sclerosis in the UK. Value Health 2007;10:54-60

We added these details on utility estimation in the supplementary material.

11. Describe the interferon beta 1b trial from which the MAIC was conducted. It seems like you used the data from the North American trial. Please justify why you used this and consider running an additional analysis using the European trial data.

The MAIC used in the model was the one published by Samjoo et al. More details on the method used are reported in that paper (Samjoo 2020). 

We have better specify in the text, the availability of MAIC methods details in the paper by Samjoo et al. Further, we have added a sentence to explain which data source was used in the MAIC: “The relative efficacy estimated with the MAIC was based on the siponimod trial data and the interferon beta-1b trial conducted by Panitch et al (Panitch 2004). The study by Panitch et al., conducted in North America, was selected because was the one that assessed the impact of treatment on disability progression using the confirmed disability progression at 6 months”. 

Samjoo IA, Worthington E, Haltner A, Cameron C, Nicholas R, Rouyrre N, Dahlke F, Adlard N. Matching-adjusted indirect treatment comparison of siponimod and other disease modifying treatments in secondary progressive multiple sclerosis. Curr Med Res Opin. 2020 Jul;36(7):1157-1166

Panitch H, Miller A, Paty D, et al. Interferon beta-1b in secondary progressive MS: results from a 3-year controlled study. Neurology. 2004;63(10):1788–1795

12. The MAIC relied on summary data from the IFN trials and had a lot of limitations. For example individual trials used different definitions of CDP (including the North American trial) and the MAIC could not adjust for all potential effect modifiers. These limitations should be addressed in the discussion of your manuscript. Given these limitations, you may also want to consider adding an additional comparator for best supportive care (estimated by the placebo arm of EXPAND).

We thank the reviewer for the comment. We have added the MAIC limitation on the discussion section.

“Even if MAIC is the best approach to estimate the relative efficacy between siponimod and interfern beta-1b, some limitations are associated to this approach. As reported by Samjoo et al., these limitations include differences in trial and patient characteristics, which were not fully adjusted due to a paucity of data. Inclusion criteria for the IFNb-1b trials were broader than EXPAND thus precluding our ability to align on all variables despite individual patients data from EXPAND. Fortunately, for the key treatment effect modifiers identified by clinical experts, multivariable adjustments in the MAIC were possible. The definition of “disability progression” reported difference between studies. Although these MAICs adjust for observed baseline differences between siponimod and comparator trials, they are comparisons of randomized treatment groups and may therefore be biased by potential unobserved cross-trial differences.” 

13. Please clarify how you estimated the clinical effectiveness in the model. I’m not sure if I am understanding Table 2 correctly but it seems like you used the relative effects vs placebo rather than the indirect estimate from the MAIC of siponimod vs. IFN?

We thank the reviewer for the comment. MAIC analyses provided an anchored indirect comparison due to the common comparator arm in each comparison (placebo) (Phillippo 2016). We used the relative effectiveness of interferon beta 1b and siponimod versus placebo in order to adjust the disability progression matrix estimated based on placebo arm of siponimod trial.

This approach was better explained in the method section 

Phillippo DM, Ades A, Dias S, et al. NICE DSU Technical Support Document 18: Methods for population-adjusted indirect comparisons in submission to NICE [Internet]. 2016. Available from: http://nicedsu.org.uk/wp-content/ uploads/2018/08/Population-adjustment-TSD-FINAL-ref-rerun.pdf

Results

14. I wonder if the incremental cost is reported correctly. The ICER I calculate using the disaggregated costs is closer to what you report than the ICER I calculate when I use the incremental values from Table 5.

We thank the reviewer for the comment. There was an error in the delta cost reported, the right difference is €30.308 and not 25,023. We have adjusted the value of the delta cost in the table.

However, the ICER reported was correct and already estimated using the 30,308 difference. Also the overall costs of the two product were correct.

Discussion

15. In the comparison to the US model, it would be better to review the full HTA report from the Institute for Clinical and Economic Review (rather than the summary in JMCP). They conducted a scenario analysis using the MAIC data to compare siponimod with IFN. You will also find a budget impact analysis in that report.

We thank the reviewer for the comment. We have added more detail on US analysis and the difference compare to our results on Italian setting.

However, no data on budget impact analysis were reported in the ICER HTA as stated in the report: “As discussed above with regard to value-based price benchmarks, the FDA-approved indication for siponimod (relapsing forms of MS) is different from the focus of this review (SPMS). As such, we are not providing calculations related to the potential budget impact of siponimod.” Based on the lack of this data with di not have the possibility to compare our budget impact results with the US one.

---

## [Decision Letter · Decision Letter 1]

10 Jan 2022

PONE-D-21-28328R1Cost-effectiveness and budget impact analysis of Siponimod in the treatment of Secondary progressive Multiple Sclerosis in ItalyPLOS ONE

Dear Dr. Antonazzo,

Thank you for submitting your manuscript to PLOS ONE. After careful consideration, we feel that it has merit but does not fully meet PLOS ONE’s publication criteria as it currently stands. Therefore, we invite you to submit a revised version of the manuscript that addresses the points raised during the review process.

We look forward to receiving your revised manuscript.

Kind regards,

Marcello Moccia

Academic Editor

PLOS ONE

Journal Requirements:

Reviewers' comments:

Reviewer's Responses to Questions

**Comments to the Author**

1. If the authors have adequately addressed your comments raised in a previous round of review and you feel that this manuscript is now acceptable for publication, you may indicate that here to bypass the “Comments to the Author” section, enter your conflict of interest statement in the “Confidential to Editor” section, and submit your "Accept" recommendation.

Reviewer #2: (No Response)

2. Is the manuscript technically sound, and do the data support the conclusions?

Reviewer #2: Yes

3. Has the statistical analysis been performed appropriately and rigorously? 

Reviewer #2: I Don't Know

4. Have the authors made all data underlying the findings in their manuscript fully available?

Reviewer #2: Yes

5. Is the manuscript presented in an intelligible fashion and written in standard English?

Reviewer #2: No

6. Review Comments to the Author

Reviewer #2: Thank you for addressing my comments. I'm afraid I still do not understand how you used the MAIC to adjust the disability progression matrix

Your rationale for using CDP-6 instead of CDP-3 makes sense, although a scenario analysis using the CDP-3 results from the MAIC/European trial would help to characterize the uncertainty in your CEA.

A few other minor comments:

-Table 1: You may want to include a citation for the dimethyl fumarate study you referenced for utilities

-Spellcheck and review the acronyms (e.g. SMPS instead of SPMS)

-The clarity and readability of the paper would benefit from editorial review by a native English speaker

7. PLOS authors have the option to publish the peer review history of their article (what does this mean?). If published, this will include your full peer review and any attached files.

Reviewer #2: No

---

## [Author Response · Author response to Decision Letter 1]

2 Feb 2022

Dear Editor, 

Thanks for the opportunity to address the reviewer’ comments, which was carefully considered when preparing the new revised version of the manuscript. 

Our point-by point responses are provided below

Reviewers' comments:

Reviewer #2: 

Thank you for addressing my comments. I'm afraid I still do not understand how you used the MAIC to adjust the disability progression matrix

We thank the reviewer for the comment. In the new version of supplemental material, we better described this aspect by including the follows sentence “The effectiveness of included DMTs was estimated performing a MAIC using data from recent literature review based on clinical trial data and EXPAND data.14 The treatment efficacy was reported as Hazard Ratio (HR) using the Confirmed Disability Progression (CDP) at 6 months and as Rate Ratio (RR) using relapse rate.15 To estimate the reduction of relapse rate and disease progression associated to each treatment, we applied the RR and HR estimated in the review to the natural history probabilities.”. 

Your rationale for using CDP-6 instead of CDP-3 makes sense, although a scenario analysis using the CDP-3 results from the MAIC/European trial would help to characterize the uncertainty in your CEA.

We thank the reviewer for the comment. In the new version of manuscript, we have added a sensitivity analysis to evaluate CDP-3 as suggested by author. Therefore, we have added the following sentence in the methods “Finally, an alternative scenario analysis based on DMTs efficacy assessed using the Confirmed disability progression at 3 months (CDP-3) was performed. The CDP-3 data was retrieved by the MAIC published by Samjoo et al.12 with a Hazard Ratio of 0.74 (95%CI: 0.60-0.91) for interferon beta-1b and 0.61 (0.32-1.16) for siponimod.”, the follows one in the results “The impact of treatment efficacy was also confirmed by the alterative scenario analysis. This analysis based on CDP-3 data, siponimod reported an ICER of € 80,063 compare to interferon beta-1b.”, and finally the follows sentence in the discussion section “These results were mostly affected by DMTs efficacy, as showed by sensitivity analysis and alternative scenario analysis.”.

A few other minor comments:

-Table 1: You may want to include a citation for the dimethyl fumarate study you referenced for utilities

Amended as suggested by reviewer

-Spellcheck and review the acronyms (e.g. SMPS instead of SPMS)

Amended as suggested by reviewer

-The clarity and readability of the paper would benefit from editorial review by a native English speaker

We thank the reviewer for the comment. We revised the manuscript as suggested by reviewer. We hope the new version of paper will be clearer and more readable.

---

## [Editor Report · Decision Letter 2]

4 Feb 2022

Cost-effectiveness and budget impact analysis of Siponimod in the treatment of Secondary progressive Multiple Sclerosis in Italy

PONE-D-21-28328R2

Dear Dr. Antonazzo,

We’re pleased to inform you that your manuscript has been judged scientifically suitable for publication and will be formally accepted for publication once it meets all outstanding technical requirements.

Kind regards,

Marcello Moccia

Academic Editor

PLOS ONE
---

## [Editor Report · Acceptance letter]

18 Feb 2022

PONE-D-21-28328R2 

Cost-effectiveness and budget impact analysis of Siponimod in the treatment of Secondary progressive Multiple Sclerosis in Italy 

Dear Dr. Antonazzo:

I'm pleased to inform you that your manuscript has been deemed suitable for publication in PLOS ONE. Congratulations! Your manuscript is now with our production department. 

Kind regards, 

on behalf of

Dr. Marcello Moccia 

Academic Editor

PLOS ONE